# DNA Methylation Signature for *JARID2*-Neurodevelopmental Syndrome

**DOI:** 10.3390/ijms23148001

**Published:** 2022-07-20

**Authors:** Eline A. Verberne, Liselot van der Laan, Sadegheh Haghshenas, Kathleen Rooney, Michael A. Levy, Mariëlle Alders, Saskia M. Maas, Sandra Jansen, Agne Lieden, Britt-Marie Anderlid, Louise Rafael-Croes, Philippe M. Campeau, Ayeshah Chaudhry, David A. Koolen, Rolph Pfundt, Anna C. E. Hurst, Frederic Tran-Mau-Them, Ange-Line Bruel, Laetitia Lambert, Bertrand Isidor, Marcel M. A. M. Mannens, Bekim Sadikovic, Peter Henneman, Mieke M. van Haelst

**Affiliations:** 1Department of Human Genetics, Amsterdam Reproduction & Development Research Institute, Amsterdam University Medical Centers, Meibergdreef 9, 1105 AZ Amsterdam, The Netherlands; e.verberne@amsterdamumc.nl (E.A.V.); l.vanderlaan@amsterdamumc.nl (L.v.d.L.); m.alders@amsterdamumc.nl (M.A.); s.m.maas@amsterdamumc.nl (S.M.M.); sandra.jansen@amsterdamumc.nl (S.J.); m.a.mannens@amsterdamumc.nl (M.M.A.M.M.); m.vanhaelst@amsterdamumc.nl (M.M.v.H.); 2Department of Pathology and Laboratory Medicine, Western University, London, ON N6A 3K7, Canada; seyyedehsadegheh.haghshenas@lhsc.on.ca (S.H.); kathleen.rooney@lhsc.on.ca (K.R.); 3Verspeeten Clinical Genome Centre, London Health Sciences Centre, London, ON N6A 5W9, Canada; michael.levy@lhsc.on.ca; 4Department of Clinical Genetics, Karolinska University Hospital, 17176 Stockholm, Sweden; agne.lieden@ki.se (A.L.); britt.marie.anderlid@ki.se (B.-M.A.); 5Department of Molecular Medicine and Surgery, Karolinska Institutet, 17176 Stockholm, Sweden; 6Department of Pediatrics, Dr. Horacio E. Oduber Hospital, Dr. Horacio E. Oduber Boulevard 1, Oranjestad, Aruba; l.rafael@hoharuba.com; 7Department of Pediatrics, University of Montreal, Montreal, QC H4A 3J1, Canada; p.campeau@umontreal.ca; 8Department of Laboratory Medicine and Genetics, Trillium Health Partners, Mississauga, ON L5B 1B8, Canada; ayeshah.chaudhry@thp.ca; 9Department of Laboratory Medicine and Pathobiology, University of Toronto, Toronto, ON M5S 1A8, Canada; 10Department of Hum Genet, Radboud Institute for Molecular Life Sciences and Donders Institute for Brain, Cognition and Behaviour, Radboud University Medical Center, 6525 GA Nijmegen, The Netherlands; david.koolen@radboudumc.nl (D.A.K.); rolph.pfundt@radboudumc.nl (R.P.); 11Department of Genetics, University of Alabama at Birmingham, Birmingham, AL 35294, USA; acehurst@uab.edu; 12UF6254 Innovation en Diagnostic Genomique des Maladies Rares, 21070 Dijon, France; frederic.tran-mau-them@u-bourgogne.fr (F.T.-M.-T.); ange-line.bruel@u-bourgogne.fr (A.-L.B.); 13Équipe Génétique des Anomalies du Développement (GAD), CHU Dijon-Bourgogne, 21000 Dijon, France; 14Service de Génétique Clinique, CHRU Nancy, 54000 Nancy, France; l.lambert@chru-nancy.fr; 15Service de génétique médicale, CHU de Nantes, 44000 Nantes, France; bertrand.isidor@chu-nantes.fr

**Keywords:** *JARID2*, developmental disorder, DNA methylation, epigenetics, episignature

## Abstract

*JARID2* (Jumonji, AT Rich Interactive Domain 2) pathogenic variants cause a neurodevelopmental syndrome, that is characterized by developmental delay, cognitive impairment, hypotonia, autistic features, behavior abnormalities and dysmorphic facial features. *JARID2* encodes a transcriptional repressor protein that regulates the activity of various histone methyltransferase complexes. However, the molecular etiology is not fully understood, and *JARID2*-neurodevelopmental syndrome may vary in its typical clinical phenotype. In addition, the detection of variants of uncertain significance (VUSs) often results in a delay of final diagnosis which could hamper the appropriate care. In this study we aim to detect a specific and sensitive DNA methylation signature for *JARID2*-neurodevelopmental syndrome. Peripheral blood DNA methylation profiles from 56 control subjects, 8 patients with (likely) pathogenic *JARID2* variants and 3 patients with *JARID2* VUSs were analyzed. DNA methylation analysis indicated a clear and robust separation between patients with (likely) pathogenic variants and controls. A binary model capable of classifying patients with the *JARID2*-neurodevelopmental syndrome was constructed on the basis of the identified episignature. Patients carrying VUSs clustered with the control group. We identified a distinct DNA methylation signature associated with *JARID2*-neurodevelopmental syndrome, establishing its utility as a biomarker for this syndrome and expanding the EpiSign diagnostic test.

## 1. Introduction

*JARID2* (Jumonji, AT Rich Interactive Domain 2; OMIM 601594) haploinsufficiency has been associated with a clinically distinct neurodevelopmental syndrome [1,2,3]. It is characterized by developmental delay, cognitive impairment (ranging from borderline intellectual functioning to severe intellectual disability), hypotonia, autistic features and behavior abnormalities. Dysmorphic facial features include high anterior hairline, broad forehead, deeply set eyes, infraorbital dark circles, depressed nasal bridge, bulbous nasal tip and full lips.

*JARID2* encodes a transcriptional repressor protein that regulates the activity of various histone methyltransferase complexes. The *JARID2* protein plays a role in the recruitment and activation of the polycomb repressive complex 2 (PRC2), which suppresses expression of target genes through histone H3 lysine 27 (H3K27) methylation [4,5]. In addition, it has been shown in mice that Jarid2 regulates Notch1 expression during cardiac development through recruitment of Setdb1, resulting in increased methylation of histone H3 lysine 9 [6]. Because of its function in epigenetic regulation, we hypothesize that *JARID2* aberrations manifest with a specific DNA methylation (DNAm) pattern. It has been demonstrated previously that disorder-specific DNAm patterns across the genome (episignatures) can be detected through EpiSign analysis [7]. With this test, 57 episignatures associated with 65 genetic syndromes can be currently detected, including ADNP syndrome, CHARGE syndrome, Down syndrome, Kleefstra syndrome 1 and Kabuki syndrome 1 and 2 [8]. Most of the episignatures previously discovered involve genes that are part of the epigenetic machinery described by Bjornsson [9].

An important, and currently applied, clinical utility of DNAm signatures involves the reclassification of previously identified variants of unknown significance (VUS) in genes linked to rare genetic disorders [10]. In addition, DNAm signatures can be used as a diagnostic tool in patients with a suspected genetic disorder and molecularly unconfirmed diagnosis [11]. Apart from these diagnostic purposes, DNAm analysis can provide insights into the molecular mechanisms underlying genetic disorders.

In this study, we aim to (1) detect a DNAm signature for the *JARID2*-neurodevelopmental syndrome using eight patients with (likely) pathogenic *JARID2* variants and (2) assess pathogenicity of three *JARID2* VUSs with the established DNAm signature.

## 2. Results

### 2.1. Identification and Assessment of an Episignature for the JARID2-Neurodevelopmental Syndrome

The clinical and molecular details of our patient cohort are summarized in Table 1 and Figure 1.

All samples within this study passed our quality control and the clean dataset involved 776314 probes. Our three-step feature selection procedure yielded 150 probes (Appendix A) with which we observed a clear clustering of patients and controls, based on hierarchical clustering and MDS visualizations. More detailed information about the selected probes is given in Appendix A. The methylation levels (β values) at those probes for Patients 1–8 and for the control samples have also been provided in Appendix A. Both of the unsupervised models indicated the presence of a robust episignature (Figure 2).

The methylation pattern of patient samples carrying *JARID2* missense VUSs (patients 9–11) were also evaluated by plotting them alongside patients with a (likely) pathogenic variant and control individuals, using the selected probes. The three *JARID2* case samples carrying VUSs clustered with the control individuals (Figure 3).

In order to assess the reproducibility of the episignature, eight rounds of “leave one out” cross-validation were performed. At each iteration, seven case samples and the matched control samples were used for probe selection, and subsequently, using the selected probes, those samples were plotted alongside the sample that was not used for probe selection, using an MDS. All testing samples clustered with the remaining case samples (Figure 4).

### 2.2. Construction of a Classification Model

In order to ensure accurate classification of *JARID2*-neurodevelopmental syndrome patients, a support vector machine (SVM) classifier was constructed using the selected set of probes. The SVM generates a methylation variant pathogenicity (MVP) score for each individual, ranging between 0–1. Higher scores indicate more similarity to the identified *JARID2*-neurodevelopmental episignature. The SVM classifier has a default cut-off of 0.5 for the MVP score to predict the class. The classifier was constructed by training the 8 case samples with (likely) pathogenic *JARID2* variants against the 56 matched control samples (used for probe selection), 75% of other control samples, and 75% of case samples from 57 other rare genetic disorders. The remaining 25% were used for testing the model. All the samples with (likely) pathogenic *JARID2* variants received MVP scores near 1, while all the control samples and case samples from other neurodevelopmental disorders received low MVP scores. Therefore, we conclude that our model has full sensitivity and specificity. The three patients with *JARID2* VUSs were also supplied into the model and received MVP scores near zero, indicating the dissimilarity of their methylation profile to the identified *JARID2*-neurodevelopmental syndrome episignature (Figure 5).

### 2.3. Differentially Methylated Regions and Gene Set Enrichment Analysis

Using the DMRcate algorithm [12] with 5% mean methylation difference and a minimum of three CpGs, no significant DMRs were detected. Additional exploration of the set of 150 selected probes yielded in total seven regions involving two or three probes with a consistent direction of effect.

### 2.4. Gene Set Enrichment Analysis

Using the different gene set enrichments web tools and distinct databases did not yield consistent associated molecular pathways.

## 3. Discussion

Generally established during embryonic development, DNAm patterns are usually detectable in peripheral blood, making them easily accessible biomarkers for use in a clinical setting [13]. DNAm episignatures have recently been used for helping to provide a more definitive diagnosis for patients with Mendelian neurodevelopmental disorders, especially the unresolved cases, as well as to reclassify VUSs [10,14,15,16,17].

In this study, we were able to detect a specific DNAm signature for *JARID2*-neurodevelopmental syndrome in a cohort of eight patients with (likely) pathogenic *JARID2* variants. Hierarchical clustering and MDS visualization showed a clear distinction between patients and controls, indicating robustness of the episignature. In order to assess the reproducibility of the episignature, we performed eight rounds of cross-validation. Each round, the testing sample clustered together with the patient samples, indicating that it is a reproducible episignature. In addition, our SVM classification model showed that the methylation signature was highly sensitive and specific for the *JARID2*-neurodevelopmental syndrome and not sensitive to other diseases that are linked to the epigenetic machinery and also associated with features such as developmental delay and intellectual disability.

The established *JARID2*-neurodevelopmental syndrome episignature was then utilized to assess the pathogenicity of the two missense variants in *JARID2*, identified in patients 9–11. These three patients clustered with the control individuals in the MDS plot and yielded MVP scores near zero, indicating their different methylation profile from that of the 8 patients with (likely) pathogenic *JARID2* variants. As episignatures may only be used to upgrade a VUS to (likely) pathogenic if the related episignature is present, these negative results do not rule out pathogenicity [18]. For example, it might be possible that *JARID2* missense variants are associated with a distinct episignature. In ADNP syndrome, for instance, two distinct episignatures have been identified, as a result of truncating variants in two distinct protein domains of the *ADNP* gene [17]. However, since clinical features of the three patients with a *JARID2* VUS are not highly specific to only *JARID2*-neurodevelopmental syndrome, we think that these *JARID2* missense variants might very well be benign variants and that the phenotype of these patients is caused by another, currently unknown, genetic aberration. This is further supported by the observation that patients 9 and 10 did not show the facial dysmorphisms that have been associated with *JARID2*-neurodevelopmental syndrome. The phenotype of patient 11 included severe global spasticity and lack of myelination on brain MRI, both not known to be associated with *JARID2*-neurodevelopmental syndrome at present. In addition, the missense variant c.2363G > A p.(Arg788Gln) that was identified in patients 9 and 11 is not located in any of the currently known domains that are important for functioning. Moreover, the amino acid change from arginine to glutamine is predicted to lead to a normal functioning protein [19]. The missense variant c.1930G > A p.(Glu644Lys) that was identified in patient 10 is located in the BRIGHT, ARID domain (AT-rich interaction). This domain plays an important role in developmental, tissue-specific gene expression and proliferation control and it can potentially bind DNA [20,21]. However, the amino acid change from glutamate to lysine is predicted to lead to a normal functioning protein [19]. No other possible disease-causing variants were identified by whole exome sequencing in these three patients, although patient 9 carried two de novo VUSs in the *TNRC18* gene, for which there are currently no known disease associations. In addition, microarray analysis was performed in patient 10, with normal results.

DMR analysis did not detect any significant *JARID2* associated region. However, when we evaluated the list of 150 selected probes, we observed seven regions that involved >2 selected probes, which were located in direct vicinity of each other and showed a consistent direction of effect. Five out of these seven regions were annotated to genes of which the function remains unclear in relation to the *JARID2* disease phenotype. However, two of these seven regions were annotated to genes previously linked with aberrant neurological development. First, the *NLGN2* gene previously was associated with autism and pervasive developmental disorders [22]. Secondly, the *ADGRL2* (*LPHN2*) gene was previously associated with a mild brain malformation (rhombencephalosynapsis) [23]. Detailed follow-up studies are needed to explore whether these association indeed play a role in disease or intellectual deficit at a functional level. Further exploration of the 150 selected probes using different gene set enrichment web tools and distinct databases did not yield consistent associated molecular pathways. However, we did detect enrichment of genes annotated to the endochondral ossification pathway (Webgestalt, Wikipathway). Whether the latter truly is related to particular dysmorphic features of the *JARID2* disease phenotype remains, however, unclear and needs further in-depth translational research. However, regardless of the underlying biology, the clear association between the detected DNA methylation signature and the *JARID2*-neurodevelopmental syndrome establishes its utility in diagnostic testing for *JARID2*-neurodevelopmental syndrome.

A possible limitation of this study is the relatively small sample size of eight patients. However, considering the rarity of Mendelian neurodevelopmental disorders, episignatures are identified, using as few as five patients [8]. The availability of more samples in the future may facilitate the identification of more sensitive and specific episignature for the *JARID2*-neurovelopmental syndrome.

## 4. Materials and Methods

### 4.1. Subjects and Study Cohort

The patient cohort included a total of eleven individuals (six males and five females) with variants in *JARID2*, of which seven (patient 1–5, 9 and 10) have been previously described in the literature [1]. All patients were identified in a clinical diagnostic setting. The *JARID2* variants had been identified through microarray analysis, whole exome sequencing or an autism/intellectual disability gene panel and were classified according to the guidelines of the American College of Medical Genetics (ACMG) [24]. Eight patients carried a pathogenic or likely pathogenic variant, including six patients with a deletion of at least one exon of *JARID2*, one patient with a frameshift variant and one with a splice site variant. Three patients carried a VUSs in the *JARID2* gene; all three were missense variants. Since haploinsufficiency of *JARID2* has been identified as the disease mechanism in *JARID2*-neurodevelopmental syndrome [1,2], the eight patients carrying a (likely) pathogenic *JARID2* variant were used for the discovery phase, i.e., episignature detection. Subsequently, the established episignature was used to further assess pathogenicity of the three missense variants. We selected 56 control samples from the EpiSign Knowledge Database (EKD), https://episign.lhsc.on.ca/index.html, matched by age, sex and array type. EKD is a database consisting of more than 600 healthy control samples and above 1000 individuals from 57 neurodevelopmental disorders with a known episignature. The samples in the EKD include samples that were processed in the same batches as those from the study cohort at AUMC and were then sent to LHSC for methylation analysis. Control samples were all selected from the same batches as our *JARID2* case samples, and none of them had a *JARID2* variant or a clinical diagnosis of *JARID2*-neurodevelopmental syndrome. The age of control individuals ranged from 2 to 50 years, with a mean of 20.1 years and a median of 10 years. In total, 46% percent of them were females and 54% males. The control to case ratio was increased until the matching quality reached an optimum point, meaning that the case and control cohorts have the highest similarity with regard to their age, sex and array type. This resulted in a control to case ratio of 7:1 [10].

### 4.2. DNA Isolation and Methylation Analysis

Peripheral blood DNA was obtained according to standard techniques. DNA methylation analysis of the samples were performed using the Illumnia Infinium methylation EPIC bead chip arrays (San Diego, CA, USA) according the manufacturer’s protocol. Data analysis was performed at the Verspeeten Clinical Genome Centre at London Health Sciences, Canada. The analysis and discovery of episignatures were carried out based on laboratory’s previously published protocols [7,10,11]. In order to minimize batch effect herein, samples were randomly divided over separate batches.

### 4.3. Quality Control of DNAm Profiles and Feature Selection

The details of the quality control and feature selection procedures were previously described in detail [25]. In brief, methylated and unmethylated signal intensities were imported into R (v 4.0.5) and normalized using background correction available under the minfi (v 1.34.0) package [26]. Probes that were annotated to the allosomes, contained single-nucleotide polymorphisms (SNPs) at or near CpG interrogation or single nucleotide extension, had detection *p*-value above 0.01 or were known to cross-react with chromosomal locations other than their target regions were eliminated from the dataset. Principal component analysis (PCA) was performed in order to detect outliers and observe the overall batch structure. Methylation level (β-value) for each probe was calculated as the ratio of methylated signal intensity to the sum of methylated and unmethylated signal intensities. In order to obtain homoscedasticity, these values were converted into M-values using the formula log2(β/(1 − β)). Next, differential methylation analysis between patients with (likely) pathogenic *JARID2* variants and controls was performed using a linear model available in the limma package (v 3.44.3). Blood cell proportions were estimated by the Houseman method [27] and included as covariates in the model matrix. The blood cell types used as covariates are CD4+ and CD8+ T-cells, natural killer cells, monocytes, granulocytes and b-cells as specified in the minfi package.

Since the Infinium EPIC bead chip array represents a high dimensional dataset (~860 K probes, raw data), we followed a three-step feature selection procedure. First, the 1000 most significant probes based on effect size and *p*-value, i.e., probes with the highest product of mean methylation differences between the case and control groups and negative of the logarithm of *p*-values, were selected. Subsequently, 250 probes with the highest areas under the receiver operating characteristic curve (AUROC) were retained. Finally, probes with a correlation higher than 0.9 within case and control groups separately were eliminated. The methylation levels at the remaining 150 probes were considered as the identifying episignature for the *JARID2*-neurodevelopmental syndrome episignature. In order to assess the robustness of the selected probes in distinguishing between the case and control groups we applied unsupervised models, herein, hierarchical clustering was performed using Ward’s method on Euclidean distance, and multidimensional scaling (MDS) was applied by measuring of the pair-wise Euclidean distances between samples. In order to guarantee the stability and reproducibility of the episignature, an eight-round leave-one-out cross-validation was performed: in each round, seven patient samples were used for probe selection, while the eighth patient sample was used as a testing sample. Subsequently, the corresponding MDSs were visualized.

### 4.4. Construction of the Binary Classifier

In order to classify case and control samples more accurately, we applied the aforementioned identified probe set in a support vector machine (SVM), which was constructed using the e1071 package, as previously described [7,10,11]. The classifier is constructed by training the 8 samples with (likely) pathogenic *JARID2* variants against the 56 matched control samples that were used for probe selection, 75% (N = 461) of other controls, and 75% (N = 939) of samples from 57 other neurodevelopmental disorders from the EKD, and the remaining 25% (N = 496) of these controls and other neurodevelopmental disorder samples were used for model testing. The model creates scores ranging 0–1 for each sample, which indicate the probability that the sample has a methylation profile similar to that of *JARID2*-neurodevelopmental syndrome. This score is called the methylation variant pathogenicity (MVP) score. By default, the SVM model uses a cut-off score of 0.5 for classifying the samples.

### 4.5. Differentially Methylated Regions

The existence of differentially methylated regions (DMRs) was investigated using the DMRcate package [12], where regions containing at least three CpGs within 1 kb with a minimum methylation difference of 5% and a Fisher’s multiple comparison *p*-value < 0.01 were considered significant. Further exploration for detection of DMRs was based on total set of 150 selected probes. Herein, DMRs were defined as including >2 differentially methylated probes in direct vicinity and showing consistent direction effect.

### 4.6. Geneset Enrichment Analysis

Geneset enrichment analysis was based on DAVID [28] and Webgestalt publicly available webtools [29], using KEGG and Wikipathway databases, according default parameters.

## 5. Conclusions

In this study, we identified a highly specific DNAm signature for patients with *JARID2*-neurodevelopmental syndrome. The signature can be used to assess and reclassify *JARID2* genomic variants. In addition, the *JARID2* signature can be added to the growing list of syndromes that can be confirmed by EpiSign analysis, thus further confirming the value of EpiSign as a diagnostic tool in patients with suspected genetic disorders.

## Figures and Tables

**Figure 1 ijms-23-08001-f001:**
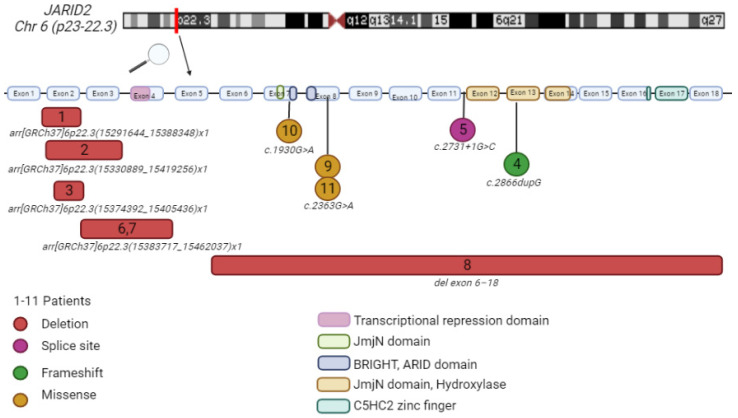
Patients’ genetic information. The numbers match with the numbers in the table and figures. Comparison between the patients with deletions (red square), splice site (purple circle), frameshift (green circle) and missense (yellow circle) variants. Patients 6 and 7 have the same deletion and are related (mother and daughter). Patient 9 and 11 are not related to each other. Alamut Visual version NM_004973.4 *JARID2*. Created with BioRender.com [1,5].

**Figure 2 ijms-23-08001-f002:**
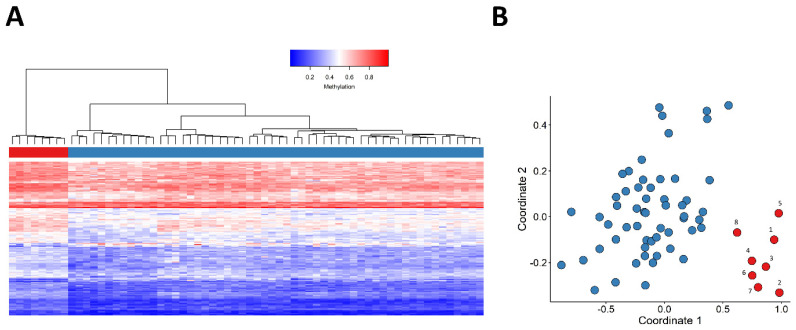
Assessment of the robustness of the *JARID2*-neurodevelopmental syndrome episignature in distinguishing between the case and control groups using unsupervised models. (**A**) Hierarchical clustering model, wherein rows represent probes and columns represent individual samples. Patients and control samples are depicted with red and blue, respectively. Heatmap gradient colors scale illustrates methylation levels ranging from blue (no methylation) to red (full methylation). (**B**) Visualization of robust segregation of patients and controls by multidimensional scaling (MDS); *X*-axis represents coordinate 1 and *Y*-axis represents coordinate 2 and red and blue circles represent case and control samples, respectively. Patients and control samples (depicted in red and blue, respectively) clearly cluster separately.

**Figure 3 ijms-23-08001-f003:**
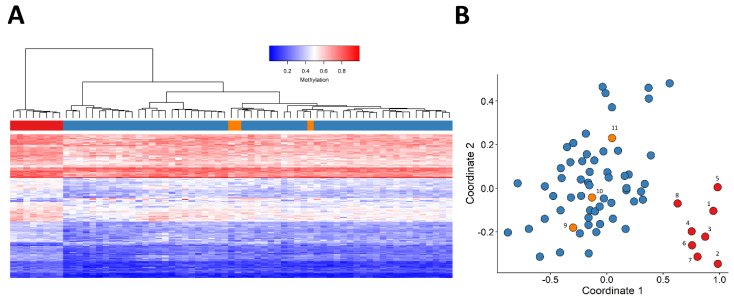
Adding patients 9−11 as a testing sample to the unsupervised clustering models. In both figures, individuals depicted in red (patients 1−8) and blue (control samples) were used for feature selection, and individuals indicated with orange (patients 9–11) were not used for selecting probes. (**A**) Hierarchical clustering, (**B**) Multidimensional scaling (MDS) plot, *X*-axis represents coordinate 1 and *Y*-axis represents coordinate 2. Patients and control samples (depicted in red and blue, respectively) clearly cluster separately. The samples with *JARID2* VUS (orange) clustered together with controls.

**Figure 4 ijms-23-08001-f004:**
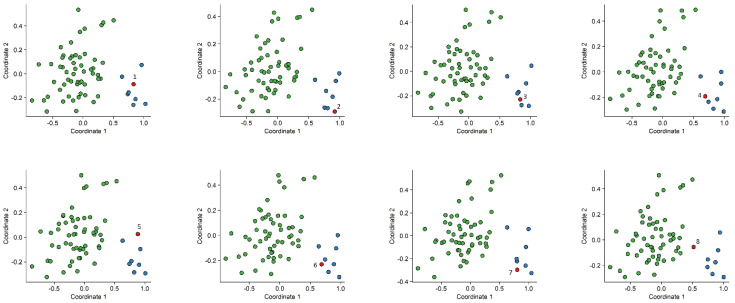
Multidimensional scaling (MDS) plots generated after eight rounds of “leave one out” cross-validation. *X*-axis represents coordinate 1 and *Y*-axis represents coordinate 2, leaving one case sample for testing (annotated in red) at each round. Control samples are annotated in green and patient samples annotated in blue. Robust clustering of the test patient with discovery patients was observed.

**Figure 5 ijms-23-08001-f005:**
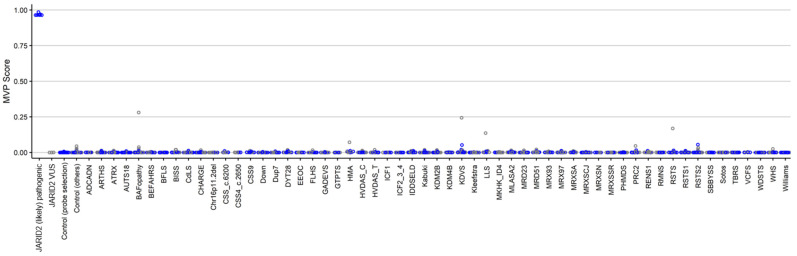
MVP scores generated by the support vector machine (SVM) classification model. All of the case samples (*JARID2* (likely) pathogenic) received high MVP scores, in contrast to the control samples and individuals from all the other disorders that received low scores, indicating the full specificity of the model. The three samples with *JARID2* VUSs yielded scores near zero.

**Table 1 ijms-23-08001-t001:** Patients’ clinical and genetic characteristics. Patient 7 is the mother of patient 6. Patient 9 and 11 are not related.

Patient #	1	2	3	4	5	6	7	8	9	10	11
**Variant**	(15291644_15388348)x1	(15330889_15419256)x1	(15374392_15405436)x1	c.2866dupG p.(Glu956GlyfsTer72)	c.2731 + 1G > C	(15383717_15462037)x1	(15383717_15462037)x1	deletion exons 6-18 *	c.2363G > A p.(Arg788Gln)	c.1930G > A p.(Glu644Lys)	c.2363G > A p.(Arg788Gln)
**Genomic position (hg19)**	-	-	-	(15511546dup)	(15507648G > C)	-	-	-	(15501555G > A)	(15497386G > A)	(15501555G > A)
**Variant type**	Del	Del	Del	FS	SS	Del	Del	Del	Mis	Mis	Mis
**Inheritance**	dn	dn	dn	dn	dn	mat	NA	dn	dn	dn	mat †
**Classification**	P	P	P	P	LP	P	P	P	VUS	VUS	VUS
**Gender**	M	F	F	M	M	F	F	M	F	M	M
**Age (years)**	14	21	18	14	5	6	36	9	43	12	8
**Intellectual functioning**	Mild ID (IQ 61-74)	Borderline intellectual functioning (IQ 82)	Mild ID (IQ 50)	Moderate ID(IQ NA)	IQ NA	Normal	Learning difficulties (IQ NA)	Mild ID (IQ NA)	Learning difficulties (IQ 79)	Mild ID (IQ 66)	Normal
**Developmental delay**	+	+	+	+	+	+	+	+	+	+	+
**Behavior abnormalities**	-	+	-	+	-	-	-	+	+	-	-
**Autistic features**	+	-	+	+	-	-	-	+	+	-	-
**ASD diagnosis**	-	-	-	+	-	-	-	+	+	-	-
**Hypotonia**	-	-	-	-	+	-	-	+ (mild)	-	-	+ (later spasticity)
**Gait disturbance**	-	-	-	-	-	+ (previously rigid walking pattern)	-	+	-	-	+ (due to spasticity)
**MRI abnormalities**	NA	NA	NA	Small posterior fossa cyst or mega cisterna magna	Arachnoid cyst	NA	NA	Normal spinal cord MRI	NA	NA	Brain MRI: lack of myelinization. Normal spinal MRI.
**Dysmorphic features**											
- Broad forehead	+	-	-	-	+	-	-	+	-	-	+
- High anterior hair line	-	+	+	+	-	+	-	+	-	-	+
- Prominent supraorbital ridges	-	-	-	-	-	-	-	+	-	-	-
- Deeply set eyes	-	+	+	-	+	-	-	+	-	-	+
- Infraorbital dark circles	+	-	+	-	-	+	-	+	-	-	+
- Midface hypoplasia	-	+	-	-	-	-	-	-	-	-	+
- Depressed nasal bridge	-	-	+	-	-	-	-	slight	-	-	-
- Bulbous nasal tip	-	-	+	-	-	+	+	-	-	-	+
- Short philtrum	-	+	+	+	-	-	-	-	-	-	+
- Full lips	-	-	+	+	-	-	-	-	-	-	+
**Other anomalies**	Pes plano valgus, mild hypermetropia	Submucous cleft palate, bifid uvula	Fetal finger pads, slight tapering of digit II and V bilateral.	2 café au lait macules	Right cryptorchidism, congenital torticollis	Supernumerary tooth	-	Kyfoscoliosis, bladder spasticity	Strabismus convergens, camptodactyly digiti V of the hands, syndactyly dig 2–3 of the feet		Severe global spasticity, neurogenic bladder

Variants based on NM_004973.4. Del; Deletion, FS; Frameshift, SS; Splice site, Mis; missense, dn; de novo, mat; maternal, LP; likely pathogenic, P; pathogenic, VUS; variant of unknown significance, F; Female, M; Male, ID; intellectual disability, ASD; autism spectrum disorder, NA; not assessed, -; absent, +; present. Patient 1, 2, 3, 4, 6 and 7 are analyzed with array techniques and the notation arr[GRCH37]6p22.3 is used. * Exact position is not known. † Mother apparently unaffected.

## Data Availability

All data generated or analyzed during this study are available from the corresponding author on reasonable request.

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
