# Peer review of "DNA Methylation Signature for JARID2-Neurodevelopmental Syndrome"

_ijms, 2022, doi:10.3390/ijms23148001_

Round 1

Reviewer 1 Report

The authors describe a methylation signature for the JARID2 associated neurodevelopmental disorder. They determined this signature based on 8 DNA samples harboring JARID2 LoF aberrations. Moreover, they did not observe this signature in three individuals with a missense variant (VUS) in this gene. These methylation signatures are important for the genetics field.

This is a well written paper that clearly describes how the signature was established. I only have some minor remarks/questions.

Figure 1. I would change 'round' to 'circle', e.g. purple circle (instead of purple round).

Table 1. Could you also add the allele frequency of the variants in gnomAD? Or, if not present in gnomAD, mention this somewhere in the text? I also prefer to include the genomic location of the single nucleotide variants (in addition to the c. and p. notation).

Line 106: I assume the following should be in the caption of Table 1 and not in the main text: '* Exact position is not known. + Mother apparently unaffected'. Please change this.

Line 106. The authors mention that the mother of individual 11 is unaffected. Was this mother deeply phenotyped?

Figure 2 & 3. Please add some explanation about the clustering of the samples in the caption, e.g. Patients and control samples (depicted in red and blue, respectively), clearly cluster seperately.

Line 156 extra white space between received and MVP scores

Line 193 extra white space between utilized & to assess.

Line 200 JARID2 missense variants (use Italics for the gene). 

Lines 208-214 What was the result of other prediction tools for these variants? Was arrayCGH also performed on the indivuals with the JARID2 missense variants? In addition, was WES analysis performed on these individuals or only targeted panel analysis? I assume no other possible causal variants were identified in these individuals? Please clearly mention this in the manuscript.

Line 254 Why were 56 control samples selected? 

General remark/suggestion: In the mean time, do the authors have an extra sample with a JARID2 LoF variant available? If yes, it might be nice to test this sample for the signature and add this to the manuscript, providing extra evidence for the signature.

Author Response

Figure 1. I would change 'round' to 'circle', e.g. purple circle (instead of purple round).

Response point 1; Thank you for the suggestion, we changed the legend of Figure 1 accordingly.

Table 1. Could you also add the allele frequency of the variants in gnomAD? Or, if not present in gnomAD, mention this somewhere in the text? I also prefer to include the genomic location of the single nucleotide variants (in addition to the c. and p. notation).

Response point 2; The genomic location of the SNVs are added.

Line 106: I assume the following should be in the caption of Table 1 and not in the main text: '* Exact position is not known. + Mother apparently unaffected'. Please change this.

Response point 3; Thank you for noticing, we now changed this.

Line 106. The authors mention that the mother of individual 11 is unaffected. Was this mother deeply phenotyped?

Response point 4; Yes, evaluation was performed by the clinical geneticist, including deep phenotyping.  

Figure 2 & 3. Please add some explanation about the clustering of the samples in the caption, e.g. Patients and control samples (depicted in red and blue, respectively), clearly cluster seperately.

Response point 5; We added some more explanation in figure 2&3

Line 156 extra white space between received and MVP scores

Response point 6; Thank you for noticing, we now changed this.

Line 193 extra white space between utilized & to assess.

Response point 7; Thank you for noticing, we now changed this.

Line 200 JARID2 missense variants (use Italics for the gene). 

Response point 8; Thank you for noticing, we now changed this.

Lines 208-214 What was the result of other prediction tools for these variants? Was arrayCGH also performed on the indivuals with the JARID2 missense variants? In addition, was WES analysis performed on these individuals or only targeted panel analysis? I assume no other possible causal variants were identified in these individuals? Please clearly mention this in the manuscript.

Response point 9; Thank you for these relevant questions. We now further clarify this in the manuscript (line 221-224).

Line 254 Why were 56 control samples selected? 

Response point 10; There were 56 control samples in order to ensure a high matching quality to the case samples. The lines 266-269 “The control to case ratio was increased until the matching quality reached an optimum point; meaning that the case and control cohorts have the highest similarity with regard to their age, sex, and array type. This resulted in a control to case ratio of 7:1.” address this comment.

General remark/suggestion: In the mean time, do the authors have an extra sample with a JARID2 LoF variant available? If yes, it might be nice to test this sample for the signature and add this to the manuscript, providing extra evidence for the signature.

Response point 11; Thank you for this suggestion. Indeed, this would provide extra evidence for the established signature, but unfortunately we currently do not have such an extra sample available.  

Reviewer 2 Report

In this study, the authors seek to identify a DNA methylation signature specific to the newly recognized neurodevelopmental disorder associated with variants in JARID2, which could then be utilized as part of the diagnostic toolkit for the practicing clinician, particularly as a supplement for cases in which a variant of uncertain significance is identified on standard genetic testing. Overall this paper is well-written, quite clear and straightforward, and will add to the utility of DNA methylation testing in clinical practice. I have outlines questions and suggestions, below, which may add to the clarity and utility of the manuscript:

1.       Regarding variant classification, what were the parameters for calling the variants (likely) pathogenic versus VUS? Is this the interpretation of the testing laboratory? Have any of these variants been identified in the prior case series for this disorder?

Perhaps my biggest concern is with overfitting. The authors used the eight samples to train the SVM, and did not rigorously test the model. Typically, these models require splitting the data into training and test sets that are independent of each other to ensure that the model is able to classify data that did not feed into the training. The test datasets used (patients 9-11) did not get classified as JARID2-neurodevelopmental signature by the model, which the authors rationalized. However, having a test data (not included in the training set) that does correctly get classified as JARID2-neurodevelopmental signature is essential to determine if this model will be widely useful.

2.       There are several examples in the prior JARID2 case reports of affected individuals with missense variants, whereas in the present study the affected individuals all had deletions, or variants inducing a frameshift or affecting a splice site, while the only missense variants were classified as VUS’s (and were subsequently not found to have the same DNAm signature as the other variants). Are there examples within EpiSign where variants in different domains or of different types (deletion versus missense, for example) result in different DNAm signatures? Related, do you have reason to believe the DNAm signature identified in your deletion/splicing/frameshift patients would be conserved in affected patients with a missense variant (and would be ‘positive’ in the previously published missense variant patients)? Or, as stated in the Discussion, would a ‘negative’ result simply mean the variant would remain as a VUS without being up- or down-graded in this case.

3.       Regarding Figure 5: The labeling is unclear. Are the (likely) pathogenic JARIDs variants the first column, and are the VUS’s the second (labelled ‘test’)?

Author Response

  1. Regarding variant classification, what were the parameters for calling the variants (likely) pathogenic versus VUS? Is this the interpretation of the testing laboratory? Have any of these variants been identified in the prior case series for this disorder?

Response point 1 – first part: Variant classification was performed by us according to the guidelines of the American College of Medical Genetics (ACMG) (line 247-249). Some of the variants had indeed been identified before, since some of the patients were previously published (line 244-246).

Perhaps my biggest concern is with overfitting. The authors used the eight samples to train the SVM, and did not rigorously test the model. Typically, these models require splitting the data into training and test sets that are independent of each other to ensure that the model is able to classify data that did not feed into the training. The test datasets used (patients 9-11) did not get classified as JARID2-neurodevelopmental signature by the model, which the authors rationalized. However, having a test data (not included in the training set) that does correctly get classified as JARID2-neurodevelopmental signature is essential to determine if this model will be widely useful.

Response point 1 – second part; We appreciate the reviewer’s concern on this matter. Given the rarity of the ¬¬JARID2- associated neurodevelopmental disorder, the optimal approach was to include all of the case samples with pathogenic JARID2 variants for probe selection and model construction. In such models the leave-one-out cross-validation can provide robust evidence for the reproducibility of the identified episignature, as has been commonly reported for other rare disorder cohorts. We do agree with the reviewer that, in the future, with additional cases identified to extend the total sample size it would be valuable to confirm the findings using an alternate approach such as training/testing cohort comparisons.

  1. There are several examples in the prior JARID2case reports of affected individuals with missense variants, whereas in the present study the affected individuals all had deletions, or variants inducing a frameshift or affecting a splice site, while the only missense variants were classified as VUS’s (and were subsequently not found to have the same DNAm signature as the other variants). Are there examples within EpiSign where variants in different domains or of different types (deletion versus missense, for example) result in different DNAm signatures? Related, do you have reason to believe the DNAm signature identified in your deletion/splicing/frameshift patients would be conserved in affected patients with a missense variant (and would be ‘positive’ in the previously published missense variant patients)? Or, as stated in the Discussion, would a ‘negative’ result simply mean the variant would remain as a VUS without being up- or down-graded in this case.

Response point 2; We would like to thank the reviewer for the insightful comment. There are examples in the literature for different episignatures for the same syndrome, based on factors such as the affected protein domain or  genotype. For instance, there are two distinct episignatures associated with Helsmoortel-Van der Aa syndrome (HVDAS), caused by variants in ADNP, depending on whether the variant involves the C- and N-terminus or the nuclear localization signal . Another example is the different methylation patterns in male and female Claes-Jensen patients, with hemizygous and heterozygous KDM5C variants . The current JARID2 signature is defined using the samples with obligate loss of function mutations. We expect that the detected episignature for the JARID2-neurodevelopmental syndrome may be able to identify patients with pathogenic missense variants with the loss of function effect. There are several rare genetic disorders with a known episignature identified using samples from patients with different variant types . For instance, a unique episignature is detected for Wiedemann-Steiner syndrome (WDSTS), where patients with missense variants present a similar methylation pattern to that of patients with truncating variants. However, not all the WDSTS patients with VUS missense KMT2A variants demonstrate the identified episignature (Foroutan et al., 2022, IJMS). As mentioned in the discussion and correctly restated by the reviewer, for a disorder with an established episignature, a VUS can be reclassified as (likely) pathogenic if the patient presents with the detected methylation profile, but it will remain a VUS otherwise We now added this to the discussion (line 201-204)

References;

(Foroutan et al., Clinical Utility of a Unique Genome-Wide DNA Methylation Signature for KMT2A-Related Syndrome, 2022, IJMS)

(Schenkel et al., Peripheral blood epi-signature of Claes-Jensen syndrome enables sensitive and specific identification of patients and healthy carriers with pathogenic mutations in KDM5C, 2018, Clinical Epigenetics)

(Schenkel et al., Identification of epigenetic signature associated with alpha thalassemia/mental retardation X-linked syndrome, 2017, Epigenetics and Chromatin

  1. Regarding Figure 5: The labeling is unclear. Are the (likely) pathogenic JARIDs variants the first column, and are the VUS’s the second (labelled ‘test’)?

Response point 3; The labels in the figure have been updated

Reviewer 3 Report

The manuscript by Verberne et al. analyzes DNA methylation in blood samples of eleven patients carrying various mutations in JARID2, three of whom carry gene variants of uncertain significance (VUS) to the pathology of JARID2 neurodevelopmental syndrome. Based on the Infinium Epic bead chip results and using a set of selection criteria the Authors identify 150 probes that show differences in the level of DNA methylation between  healthy controls and 9 patients with confirmed pathological JARID2 variants. The same analysis clearly shows that the  3 VUS variants group with  controls and are, therefore, not directly related to the pathology of this disease. 

The manuscript is concise, well written and brings novel, valuable results that may help in diagnosis of the JARID2 neurodevelopmental syndrome . 

Author Response

There where no points from this reviewer. Thank you for the compliments about our paper. 

Round 2

Reviewer 2 Report

The authors addressed my major concern and explained the choice to use the "leave-one-out" method based on the rarity of the disorder. I appreciate the difficulty in doing so and think they have done a good job in this endeavor.